# Receptor-Loaded Virion Endangers GPCR Signaling: Mechanistic Exploration of SARS-CoV-2 Infections and Pharmacological Implications

**DOI:** 10.3390/ijms222010963

**Published:** 2021-10-11

**Authors:** Qiangmin Zhang, Peter A. Friedman

**Affiliations:** Laboratory for GPCR Biology, Department of Pharmacology and Chemical Biology, University of Pittsburgh, Pittsburgh, PA 15261, USA; paf10@pitt.edu

**Keywords:** SARS-CoV-2, PDZ interaction, viral cell entry, GPCR signaling, drug development

## Abstract

SARS-CoV-2 exploits the respiratory tract epithelium including lungs as the primary entry point and reaches other organs through hematogenous expansion, consequently causing multiorgan injury. Viral E protein interacts with cell junction-associated proteins PALS1 or ZO-1 to gain massive penetration by disrupting the inter-epithelial barrier. Conversely, receptor-mediated viral invasion ensures limited but targeted infections in multiple organs. The ACE2 receptor represents the major virion loading site by virtue of its wide tissue distribution as demonstrated in highly susceptible lung, intestine, and kidney. In brain, NRP1 mediates viral endocytosis in a similar manner to ACE2. Prominently, PDZ interaction involves the entire viral loading process either outside or inside the host cells, whereas E, ACE2, and NRP1 provide the PDZ binding motif required for interacting with PDZ domain-containing proteins PALS1, ZO-1, and NHERF1, respectively. Hijacking NHERF1 and β-arrestin by virion loading may impair specific sensory GPCR signalosome assembling and cause disordered cellular responses such as loss of smell and taste. PDZ interaction enhances SARS-CoV-2 invasion by supporting viral receptor membrane residence, implying that the disruption of these interactions could diminish SARS-CoV-2 infections and be another therapeutic strategy against COVID-19 along with antibody therapy. GPCR-targeted drugs are likely to alleviate pathogenic symptoms-associated with SARS-CoV-2 infection.

## 1. Introduction

With over four million confirmed deaths as of August 2021 [1], the COVID-19 pandemic caused by SARS-CoV-2 poses a serious global health crisis. The causative virus has a linear, positive-sense, single-stranded RNA genome sharing highly similar gene sequences with SARS-CoV that caused the 2003 SARS outbreaks [2]. The clinical presentation of SARS-CoV-2 patients ranges from asymptomatic or mild to severe pneumonia-like symptoms and mortality [3]. 

The SARS-CoV-2 genome encodes 16 non-structural proteins (NSPs) such as 3CL protease NSP5, several accessory proteins, for example, ORF3a [4] and four structural proteins including spike (S), envelope (E), membrane (M), and nucleoprotein (N) shared by all coronaviruses [5]. As in the case of SARS-CoV [6], ACE2 has been identified as the receptor for entry into respiratory epithelial cells in SARS-CoV-2 infections [7]. During viral infection, the trimeric S glycoprotein is cleaved into S1 and S2 subunits, and the resulting S1 containing the receptor binding domain (RBD) is responsible for binding to ACE2 and S2 for membrane fusion [8]. Extensive vaccination strategies have been developed to introduce neutralizing antibodies mostly targeting the S protein to prevent ACE2-mediated viral invasion [9]. Rapidly developed vaccines, particularly mRNA-based COVID-19 vaccine inducing S protein-specific neutralizing antibodies, have achieved substantial success in countering the spread of SARS-CoV-2 [10]. However, in addition to vaccines, alternative therapeutics as in the case of Tamiflu (oseltamivir carboxylate) for treating influenza [11] or Molnupiravir (MK-4482, EIDD-2801) for COVID-19 [12], need to be developed to contain or mitigate the pandemic, especially considering the urgent concerns of the immune escape caused by emerging and widespread SARS-CoV-2 variants [13,14]. Herein, we discuss the functional role of the PDZ interaction in mediating SARS-CoV-2 invasion, the potentially impaired GPCR signaling and the related pharmacological implications. Comprehensive characterization of SARS-CoV-2 biology, cell signaling, and pathophysiology will provide significant clues for developing novel therapeutics to treat SARS-CoV-2 infection in humans.

## 2. PDZ Interaction Enhances SARS-CoV-2 Invasion by Paracellular Leaky Route and Receptor-Mediated Endocytosis

Accumulating evidence indicates that SARS-CoV-2 penetrates host cells through distinct transcellular and paracellular pathways (Figure 1). In the transcellular route, viral S protein binds ACE2 [7] to gain cell entry in lung [15], intestine (Figure 1A) [16], and kidney (Figure 1B,C) [17]. In the brain, NRP1 has been identified as the entry receptor to facilitate the virion internalization (Figure 1D) [18,19]. Alternatively, paracellular leak caused by epithelial barrier disorganization also facilitates SARS-CoV-2 penetration in the lung [20,21], intestine [22], and possibly in the kidney [23] and brain [24]. Notably, internalized virions and those from leaky penetration end up in the circulation by a mixed and relayed delivery combining receptor loading and paracellular leak. For instance, ACE2 mediates viral entry into the host cells and the progeny virions are generated that can be released and pass through the inter-endothelial space to blood vessels or receive ACE2 reloading and bud from the endothelial cells to the same destination (Figure 1A,B). Circulating virions subsequently reach other organs by following a hematogenous route and cause multiple organ failure as reported recently [25]. 

Remarkably, several proteins critical for viral invasion through either paracellular or transcellular route possess a canonical, carboxy-terminal PDZ-recognition sequence. Named for the structural domain shared by the postsynaptic density protein (PSD95), Drosophila disc large tumor suppressor (DlgA), and ZO-1, PDZ domains typically bind a target motif of 3-to-4 residues in length at the carboxy-tail of host proteins. These short linear motifs with certain consensus sequences are presented in three different classes: Type I, S/T-X-Φ; Type II, Φ-X-Φ; and Type III, D/E-X-Φ, where X represents any amino acid and Φ is a hydrophobic amino acid residue [28], as shown in Table 1. On the extracellular side, SARS-CoV-2 E protein containing a Type II PDZ binding motif DLLV that can interact with the PDZ domain-containing proteins PALS1 [20] or ZO-1 [23]. Utilization of cell junction-associated proteins PALS1 or ZO-1 by forming PDZ interactions disrupts the intercellular barrier and as a consequence, increases paracellular permeability facilitating viral penetration into the interstitial space and providing a mechanism through which it can gain entry to the circulation [20,23,29]. SARS-CoV-2 E protein is a short integral membrane of 75 residues embedded in the viral envelope (Figure 2A), raising the question of how it interacts with PALS1 or ZO-1, especially considering that the long stalk of viral S protein presents a physical obstacle for E reaching into the hydrophobic pocket of PDZ domains. Furthermore, the orientation of E remains unknown. More experimental evidence is needed to functionally define the role of E in mediating viral invasion compared to the well-characterized S protein. For this to end, it is required to obtain the SARS-CoV-2 spike and envelope doubly-pseudotyped virus, where E remains intact, and S has defective mutations causing its inability to interact with the ACE2 receptor. 

On the intracellular side, viral receptors ACE2 and NRP1 harbor C-terminal Type I PDZ ligands, -QTSF, and -YSEA, respectively. We recently found that the cytoplasmic PDZ domain-containing protein NHERF1 interacts with ACE2, which further aids in assembling the ACE2, B^0^AT1, NHERF1, and β-arrestin super-complex that attached to SARS-CoV-2 virions by ACE2-mediated recognition of the viral S protein. The super-complex enhances SARS-CoV-2 invasion, and the related scenario may be common in lung, intestine, and kidney, all of which express NHERF1 [35] and even recur in NRP1-mediated viral entry. Herein, the SARS-CoV-2 virion acts as a ligand that specifically binds to ACE2 or NRP1 receptors and mediates viral transcytosis. Receptor-plus-ligand-based viral entry often requires virion ligand binding to a specific receptor such as the phosphate housekeeping transporter PiT1 for leukemia virus [36], sodium/bile acid cotransporter for hepatitis B/D virus [37], or G protein-coupled chemokine coreceptors CCR5 and CXCR4 for human immunodeficiency virus (HIV-1) [38]. In the case of SARS-CoV-2, both identified receptors are single-pass membrane proteins that require the coreceptor B^0^AT1 for ACE2 [8,15] and likely VEGF-A for NRP1 [39] to maintain stable membrane residence that better supports virion loading. Additionally, SARS-CoV-2 N and 3a protein remain intracellular during viral replication and also have PDZ-binding motifs [40], -STQA, and -SVPL, respectively (Table 1). The host PDZ proteins interacting with these two proteins are not characterized and their roles remain to be ascertained. Given N protein contains a specific viral RNA packing signal [41], the unidentified cellular PDZ protein for N may serve as a docking site, scaffolding N protein-mediated genome assembly into the complete progeny virions.

## 3. Possible Damaged GPCR Signaling by SARS-CoV-2 Invasion

Upon infection, the innate immune response is triggered as the initial line of defense [42], which further prompts activation of the adaptive immune responses to provide long-lasting resistance against SARS-CoV-2 [43,44]. Except for the pathological immune responses, SARS-CoV-2 infection in various target cells impairs some normal cellular physiological functions by conscripting intercellular or intracellular signaling and causing illness. Typical symptoms of infected patients often include fever or chills, cough, shortness of breath or difficult breathing, headache, fatigue, muscle or body aches, diarrhea, and loss of smell and taste, as described previously [45]. 

As above-mentioned, the PDZ interaction between viral E protein and host PALS1 [46,47] or ZO-1 [23] could demolish cell tight junctions and, consequently, interfere with intercellular signaling and communication. Accordingly, the neighboring cells will lose the direct exchange of metabolites and signaling molecules such as extracellular messengers and hormones required for ensuing intracellular signaling transduction, as in the case of the functional definition of ATP and serotonin in paracrine signaling [48].

In addition to interrupted intercellular signaling, the noted symptoms are also relevant to dysregulation of physiological activities by specific GPCRs such as for pain, loss of smell and taste, and diarrhea. Viruses could directly utilize and impair GPCR-related intracellular signaling, which has been reported in the case of human JC polyomavirus loaded by the serotonergic receptor 5HT_2A_R [49] and HIV-1 by G protein-coupled chemokine coreceptors CCR5 and CXCR4 [38]. Instead, SARS-CoV-2 employs ACE2 or NRP1 rather than GPCRs to gain cell entry. As stated in our recent work [35], SARS-CoV-2 may hijack components such as NHERF1 and β-arrestin from various GPCR assemblies that, in turn, indirectly damage specific GPCR signaling (Figure 2B). Furthermore, β-arrestin can activate G protein-independent signaling and regulate multiple cellular signaling pathways including MAPK signaling that supports cell growth and survival [50]. Indeed, affected p38 MAPK signaling has been reported upon SARS-CoV infection [51], where its genomic sequence is highly similar to SARS-CoV-2. In the case of SARS-CoV-2, the infected cells are desensitized but appear normal in the presence of agonist or antagonist due to the lack of sufficient virion-occupied component proteins like NHERF1 and β-arrestin to assemble the GPCR signalosome, further protecting the targeted cells from being attacked by the human immune system. Similar strategies have been exploited by herpesviruses that encode their own GPCR to hijack host cell GPCR signaling networks for survival [52]. For pain responses, various well-established drug targets such as G protein-coupled opioid receptors [53] are evident candidates for investigation (Table 1). In terms of smell and taste, the olfactory receptor (OR) and taste receptor (TASR) belonging to the GPCR superfamily are the responsible sensory receptors [54]. These two receptors, widely expressed in SARS-CoV-2-susceptible lung, intestine, kidney, and brain, merit special attention. Regarding diarrhea, CFTR is of particular interest considering its role in mediating transepithelial fluid secretion primarily in the lung and intestine. CFTR is not a GPCR protein but a member of the ABC (ATP-binding cassette) transporter superfamily, often functionally coupled with and tightly controlled by certain GPCRs like the β2AR [55]. Notably, these GPCRs or non-GPCRs potentially affected by SARS-CoV-2 infection all possess C-terminal PDZ-recognition sequences, particularly NHERF1-favored Type I motif (Table 1). Hijacking NHERF1 and β-arrestin employed by these receptors damages host cellular signaling and contribute to viral pathogenesis. Indeed, a recent report suggests that upon SARS-CoV-2 infection, GPCR signaling mediated by A_2B_AR and β2AR deleteriously modulates the activity of CFTR that in turn initiates pathophysiological lung edema [56].

Collectively, viral infection directly or indirectly impairs GPCR signaling as discussed above. On the other hand, reverse regulation of GPCR signaling driven by virus invasion could seemingly reduce viral infection as in the case of the influenza virus [57], suggesting these two events are highly correlated if, not in fact, coupled.

**Table 2 ijms-22-10963-t002:** PDB structures considered in this work.

Protein ^a^	UniProt	Length	PDB ID	References
3a	P0DTC3	275	6XDC	RCSB [58]
E	P0DTC4	75	QHD43418	ZhangLab [59]
M	P0DTC5	222	QHD43419	ZhangLab
N	P0DTC9	419	QHD43423	ZhangLab
S	P0DTC2	1273	QHD43416	ZhangLab
ACE2	Q9BYF1	805	AF-Q9BYF1-F1-model_v1	AlphaFold [60]
NRP1	O14786	923	AF-O14786-F1-model_v1	AlphaFold
PALS1	Q8N3R9	675	AF-Q8N3R9-F1-model_v1	AlphaFold
ZO-1	Q07157	1748	AF-Q07157-F1-model_v1	AlphaFold
NHERF1	O14745	358	AF-O14745-F1-model_v1	AlphaFold
β-arrestin1	P49407	418	AF-P49407-F1-model_v1	AlphaFold
β-arrestin2	P32121	409	AF-P32121-F1-model_v1	AlphaFold

^a^ All full-length protein structures are from experimental or computational models as referenced. Cartoon presentations of these protein structures were prepared with PyMOL [61].

## 4. Implications in Specific Drug Development

The mechanistic exploration of SARS-CoV-2 infection holds the opportunity to discover new antiviral therapeutics. Terminating the destructive effect on lung and intestine cell tight junctions can be accomplished by preventing the interaction of PDZ formation with viral E protein and host extracellular PALS1 or ZO-1. Similarly, PDZ interactions mediated by intracellular ACE2/NHERF1 may be inhibited dramatically to reduce viral penetration. Indeed, drugs specifically targeting NHERF1 PDZ domains have been reported [62,63] and merit testing in the case of COVID-19. The fact that a G protein-coupled α2AR agonist inhibits influenza virus replication [57] underscores the potential for viruses to damage GPCR signaling. This mechanism could be harnessed to employ a broad catalog of existing approved drugs and to develop new agents against SARS-CoV-2 infection. Considering the potential side effects [64], regulating selective GPCR signaling may be particularly pertinent to alleviate or eliminate COVID-19 symptoms in affected patients. 

## 5. Concluding Remarks

Enhancing immunity by vaccination is the priority tactic stemming SARS-CoV-2 infection. Novel supplementary strategies are required for treating infected patients. For instance, disrupting extracellular or intracellular PDZ interaction may reduce viral insinuation in the targeted cells. This is of particular importance considering the potential for loading massive amounts of viruses by the paracellular route, where increased penetration across the intestine and lung may cause multi-organ infection by the vascular delivery of SARS-CoV-2. Decreased cellular virion penetration by inhibiting intracellular ACE2/NHERF1 has the potential to minimize the damage to specific target organs. Furthermore, PDZ drugs hold the promise of being more effective when combined with antibody therapy. 

Additionally, the regulation of certain GPCR signaling is also a worthy strategy to explore. Very few publications involving GPCR signaling induced by viral infection are currently available. The biggest challenge is to find an appropriate approach to evaluate GPCR signaling in specific infected model cells, particularly in the presence of authentic viruses. Notably, SARS-CoV-2-stressed GPCR signaling is a novel but ignored field, where investigating the relationship between viral infection and cell signaling is warranted. The related research may provide further optional or adjuvant medications for COVID-19 including repurposed GPCR-targeted drugs currently available and enable preparation for emerging variants of SARS-CoV-2.

## Figures and Tables

**Figure 1 ijms-22-10963-f001:**
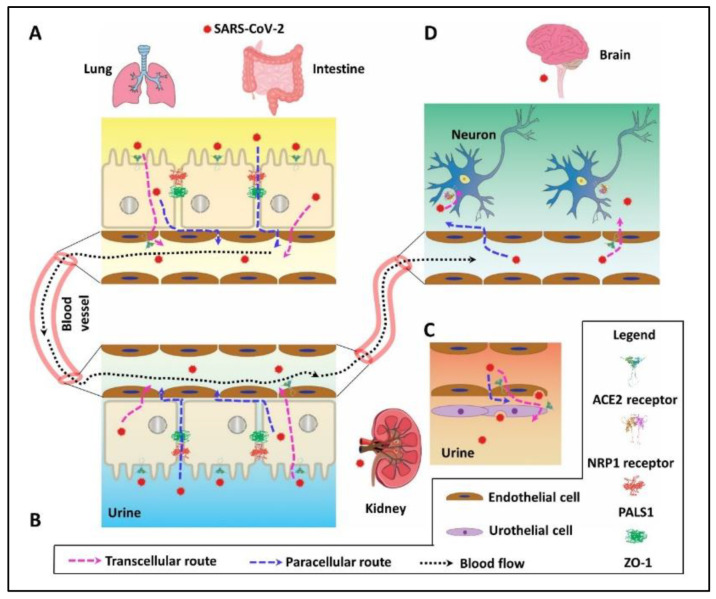
Invasive pathway of SARS-CoV-2 in the lung, intestine, kidney, and brain. (**A**) In the lung and intestine, SARS-CoV-2 may cross the epithelium barrier by ACE2 receptor-mediated endocytosis or paracellular leaky pathway. The virion can then enter the bloodstream by the paracellular leak route or ACE2-mediated transcytosis in vascular endothelial cells. (**B**) In the kidney, SARS-CoV-2 detected in urine [26] may cause acute kidney injury (AKI) [17] by ACE2-mediated renal reabsorption or paracellular leak. (**C**) SARS-CoV-2 in the urine may come from urothelial cells that receive the released virions from vascular endothelial cells [27] through ACE2-mediated transport. (**D**) In the brain, SARS-CoV-2 from the bloodstream can pass through the intercellular space or be internalized into vascular endothelial cells by ACE2-mediated endocytosis. The virions across the vascular blood–brain barrier subsequently infect brain neurons by NRP1-mediated transcytosis.

**Figure 2 ijms-22-10963-f002:**
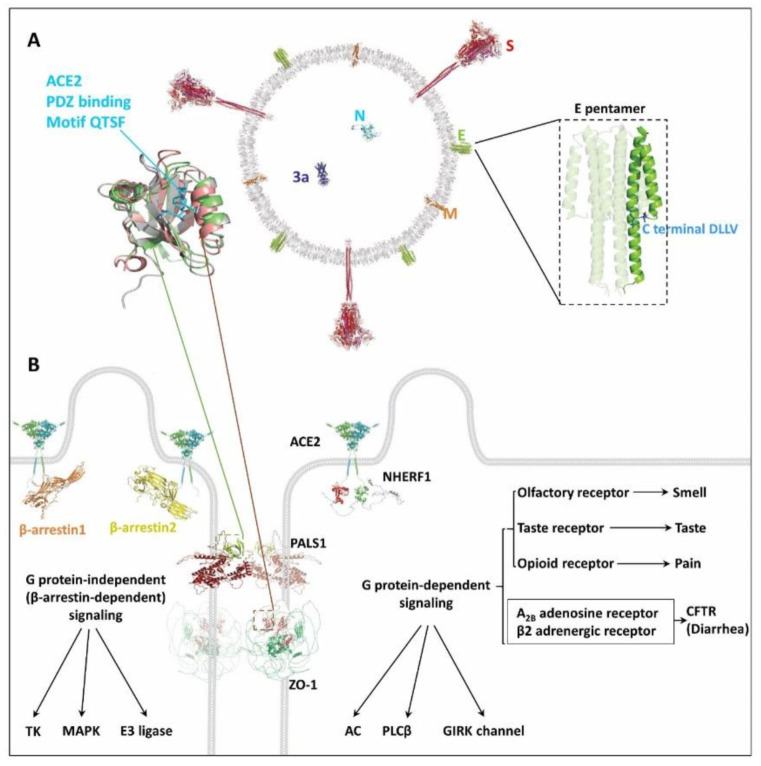
SARS-CoV-2 infection endangers GPCR signaling. (**A**) Intracellular or extracellular PDZ interaction enhances SARS-CoV-2 cell entry. SARS-CoV-2 contains viral proteins S, E, M, N, and ORF3a (Top middle panel). The virion ligand binds the ACE2 receptor and is internalized with the help of NHERF1 and β-arrestin. The virion can also cross the cellular barrier through leaky paracellular pathways. NHERF1 (gray), PALS1 (lime), and ZO-1 (salmon) have a similar β-sandwich-like PDZ domain containing six β-strands flanked by two α-helices (top left panel), where the characteristic hydrophobic pocket can accommodate the PDZ binding motif such as -QTSF (top left panel, cyan sticks) from ACE2 or -DLLV in E (top right panel, light blue sticks). The pentameric structure of full-length E is built using the available envelope protein model determined by solid-state NMR (PDB code: 7K3G) [30]. PDB identification codes (ID) for all protein structures used in this figure are listed in Table 2. (**B**) Possible damaged GPCR signaling by SARS-CoV-2 infection. SARS-CoV-2 hijacks the component proteins of the GPCR signaling machinery including NHERF1 and β-arrestins. Insufficient NHERF1 for some specific GPCRs may cause dysfunctional NHERF1-related G protein-dependent signaling [31,32] that includes the activity of AC, PLCβ, or GIRK channel [33] (lower right panel). Non-selective preference for β-arrestins hijacked by SARS-CoV-2 cell entry may cause uncontrolled β-arrestin-dependent signaling (lower left panel) that affects the activity of TK, MAPK, and E3 ligases [34], which consequently modifies downstream signaling.

**Table 1 ijms-22-10963-t001:** GPCRs and non-GPCRs containing various PDZ binding motifs.

Protein ^a^		UniProt	C-Tail Sequence	Type		Protein	UniProt	C-Tail Sequence	Type
Smell GPCR	OR5F1	O95221	SSFL	I	GPCR	A_2B_AR	P29275	GVGL	II
	OR5M10	Q6IEU7	KIAV	II		β2AR	P07550	DSLL	I
	OR10A7	Q8NGE5	LDVF	III	Non-GPCR	CFTR	P13569	DTRL	I
Taste GPCR	TAS2R20	P59543	QSTP	I		ACE2	Q9BYF1	QTSF	I
	TAS2R39	P59534	EWTL	II		NRP1	O14786	YSEA	I
	TAS2R42	Q7RTR8	ALPL	II		N	P0DTC9	STQA	I
Pain GPCR	MOR-1	P35372	APLP	II		E	P0DTC4	DLLV	II
	MOR	C7E9I8	ETAP	I		ORF3a	P0DTC3	SVPL	II

^a^ Selected GPCRs related to smell, taste, and pain, proteins harboring PDZ-binding ligand regulating CFTR, associated with secretary diarrhea, or related to SARS-CoV-2. GPCR types are shaded with different colors.

## Data Availability

The data that support the findings of this study are available from the corresponding author upon reasonable request.

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
