# Peer review of "Receptor-Loaded Virion Endangers GPCR Signaling: Mechanistic Exploration of SARS-CoV-2 Infections and Pharmacological Implications"

_ijms, 2021, doi:10.3390/ijms222010963_

Round 1
Reviewer 1 Report
Major Comments
- The purpose and scope of the review is not adequately established in the title, abstract or introduction.
- The review often lacks focus and direction and connection between ideas, concepts are not appropriately linked to virological or biological relevance. i.e. the interaction on a molecular level are not appropriately translated to virology or wider pathological effect. The review largely consists on listing all molecular interactions rather than describing there significance.
- The background information on interaction at the molecular level are not explained or discussed appropriately. The language is often imprecise and not correctly described.
- Very little virological background is provided other than molecular interactions.
- There are numerous grammatical and spelling errors.
Minor Comments:
- Line 2 - in title ‘Signaling’
- Line 28 – reference for deaths
- Line 9 – Typo ‘reach’
- Line 9 - The lung is not the primary entry point, nasal respiratory epithelium are, supported by numerous publications
- Line 31 – are therapeutics likely to ‘contain’ the pandemic or more likely reduce the disease severity in some instances – i.e. Tamiflu for influenza
- Line 39 typo - In transcellular route
- Line 42 typo - In brain,
- Line 44 – reword – gramma
Author Response
Reviewer #1: Major comments
- The purpose and scope of the review is not adequately established in the title, abstract or introduction.
Response: Rephrased (p. 2):
Herein, we discuss the functional role of the PDZ interaction in mediating SARS-CoV-2 invasion, the potentially impaired GPCR signaling and the related pharmacological implications. Comprehensive characterization of SARS-CoV-2 biology, cell signaling, and pathophysiology will provide significant clues for developing novel therapeutics to treat SARS-CoV-2 infection in humans.
- The review often lacks focus and direction and connection between ideas, concepts are not appropriately linked to virological or biological relevance. i.e. the interaction on a molecular level are not appropriately translated to virology or wider pathological effect. The review largely consists on listing all molecular interactions rather than describing the significance.
Response: The manuscript outlines the invasion route of SARS-CoV-2 and the related proteins in mediating viral cell entry. Remarkably, several proteins critical for viral invasion all involve the PDZ interaction that supports the paracellular or transcellular viral penetration. We then focus on the possible impaired GPCR signaling due to viral load burden for instance, altered GPCR signaling deleteriously modulates CFTR channeling activity that initiates a pathophysiological cascade leading to lung edema (Am J Physiol Lung Cell Mol Physiol 320: L430–L435, 2021.) as described on p. 5.
Indeed, a recent report suggests that upon SARS-CoV-2 infection GPCR signaling mediated by A2BAR and β2AR deleteriously modulates the activity of CFTR that in turn initiates pathophysiological lung edema [56].
- The background information on interaction at the molecular level are not explained or discussed appropriately. The language is often imprecise and not correctly described.
Response: now included (p.1).
The SARS-CoV-2 genome encodes 16 non-structural proteins (NSPs) such as 3CL protease NSP5, several accessory proteins for example ORF3a [4], and four structural proteins including spike (S), envelope (E), membrane (M) and nucleoprotein (N) shared by all coronaviruses [5]. As in the case of SARS-CoV [6], ACE2 has been identified as the receptor for entry into respiratory epithelial cells in SARS-CoV-2 infections [7]. During viral infection, the trimeric S glycoprotein is cleaved into S1 and S2 subunits, and the resulting S1 containing the receptor binding domain (RBD) is responsible for binding to ACE2 and S2 for membrane fusion [8].
- Very little virological background is provided other than molecular interactions
Response: The virological information of SARS-CoV-2 including its genome and proteome has now been added to the Introduction along with cited references (p.1).
The causative virus has a linear, positive-sense, single-stranded RNA genome sharing highly similar gene sequences with SARS-CoV that caused the 2003 SARS outbreaks [2]. The clinical presentation of SARS-CoV-2 patients ranges from asymptomatic or mild to severe pneumonia-like symptoms and even mortality [3].
The SARS-CoV-2 genome encodes 16 non-structural proteins (NSPs) like 3CL protease NSP5, several accessory proteins for example ORF3a [4], and four structural proteins including spike (S), envelope (E), membrane (M) and nucleoprotein (N) shared by all coronaviruses [5]. As in the case of SARS-CoV [6], ACE2 has been identified as the receptor for entry into lung epithelial cells in SARS-CoV-2 infections [7]. During viral infection, the trimeric S glycoprotein is cleaved into S1 and S2 subunits, and the resulting S1 containing the receptor binding domain (RBD) is responsible for binding to ACE2 and S2 for membrane fusion [8]. Extensive vaccination strategies have been developed to introduce neutralizing antibodies mostly targeting the S protein to prevent ACE2-mediated viral invasion [9]
- There are numerous grammatical and spelling errors.
Response: Fixed.
Minor comments:
- Line 2 – in title “signaling
Response: Some papers use “signalling”. We would like to keep it as is (See “Structure and dynamics of GPCR signaling complexes. 2018 Nature Structural & Molecular Biology 25: 4–12”)
- Line 28 –reference for deaths
Response: Cited as shown on p. 1.
With over 4 million confirmed deaths as of August 2021 [1]
- Line 9 –Typo’reach
Response: Corrected as “reaches”. Thanks very much for pointing out (p. 1).
SARS-CoV-2 exploits the respiratory tract epithelium including lungs as the primary entry point and reaches other organs through hematogenous expansion
- Line 9 - The lung is not the primary entry point, nasal respiratory epithelium are, supported by numerous publications
Response: Corrected as suggested (see Arterioscler Thromb Vasc Biol. 2020. 40:2586-2597, p. 1.). This reference is unfortunately, not cited due to the journal format issue (no reference in the abstract).
SARS-CoV-2 exploits the respiratory tract epithelium including lung as the primary entry point and reaches other organs through hematogenous expansion
- Line 31 – are therapeutics likely to ‘contain’ the pandemic or more likely reduce the disease severity in some instances – i.e. Tamiflu for influenza
Response: rephrased (p. 2).
However, in addition to vaccines, alternative therapeutics as in the case of Tamiflu (oseltamivir carboxylate) for treating influenza [11] or Molnupiravir (MK-4482, EIDD-2801) for COVID-19 [12], need to be developed to contain or mitigate the pandemic especially considering the urgent concerns of the immune escape caused by emerging and widespread SARS-CoV-2 variants [13, 14].
- Line 39 typo – In transcellular route
Response: corrected as “ In the transcellular route”. Thanks a lot.
- Line 42 typo – In brain
Response: I would like to keep it as is.
- Line 44 –reword-gramma
Response: We have run grammar check.
Reviewer 2 Report
This is a very interesting article. I would just change one thing, in my opinion the concluding paragraph is too big and should be restructured in more than one paragraph.
It is very well written providing point to point informations.
The bibliography is well selected, very consistent and relevant.
Author Response
Reviewer #2:
This is a very interesting article. I would just change one thing, in my opinion the concluding paragraph is too big and should be restructured in more than one paragraph.
It is very well written providing point to point information.
The bibliography is well selected, very consistent and relevant.
Response: Thanks very much for your nice comments and great suggestions. The concluding paragraph has now be restructured in two paragraphs.
Round 2
Reviewer 1 Report
The authour's have addressed most of the comments made.